# A Module-Level Polygenic Risk Score-Based NetWAS Framework for Identifying AD Genetic Modules Mediated by Amygdala: An ADNI Study

**DOI:** 10.3390/ijms26136060

**Published:** 2025-06-24

**Authors:** Haoran Luo, Shaoheng Fan, Hongwei Liu, Wei Li, Zhoujie Fan, Xuancheng Zhu, Chen Jason Zhang, Hong Liang, Shan Cong, Xiaohui Yao

**Affiliations:** 1Department of Computing, School of Hotel and Tourism Management, Hong Kong Polytechnic University, Hong Kong, China; haoran.luo@polyu.edu.hk (H.L.); sf1e22@soton.ac.uk (S.F.); jason-c.zhang@polyu.edu.hk (C.J.Z.); 2College of Intelligent Systems Science and Engineering, Harbin Engineering University, Harbin 150001, China; hongwei.liu@hrbeu.edu.cn (H.L.); zhoujie.fan@hrbeu.edu.cn (Z.F.); lh@hrbeu.edu.cn (H.L.); 3Qingdao Innovation and Development Center, Harbin Engineering University, Qingdao 266000, China; liwei23337@hrbeu.edu.cn (W.L.); xuanchengzhu@hrbeu.edu.cn (X.Z.)

**Keywords:** module-level PRS, network-based GWAS, mediation analysis, Alzheimer’s disease, network propagation, amygdala, MCI-to-AD conversion

## Abstract

Network-based GWAS (NetWAS) has advanced brain imaging research by identifying genetic modules associated with brain alterations. However, how imaging risk genes exert functions in brain diseases, particularly their mediation through imaging quantitative traits (iQTs), remains underexplored. We propose a module-level polygenic risk score (MPRS)-based NetWAS framework to uncover genetic modules associated with Alzheimer’s disease (AD) through the mediation of an iQT, using amygdala density as a case study. Our framework integrates genotype data, brain imaging phenotypes, clinical diagnosis of AD, and protein–protein interaction (PPI) networks to identify AD-relevant modules (ADMs) influenced by iQT-associated genetic variants. Specifically, we conducted a genome-wide association study (GWAS) of amygdala density (N=1515) to identify variants associated with iQT. These variants were mapped onto a PPI network and network propagation was performed to prompt amygdala modules. The meta-GWAS of AD ($N_{1}=63,\;926$; $N_{2}=455,\;267$) was used to calculate MPRS to further identify AD-relevant modules (ADMs). Four modules that showed significant differences in MPRS between AD and controls were identified as ADM. Post-hoc analyses revealed that these ADMs demonstrated strong modularity, showed increased sensitivity to early stages of AD, and significantly mediated the link between ADMs and AD progression through the amygdala. Furthermore, these modules exhibited high tissue specificity within the amygdala and were enriched in AD-related biological pathways. Our MPRS-based framework bridges genetics, intermediate traits, and clinical outcomes and can be adapted for broader biomedical applications.

## 1. Introduction

Genome-wide association studies (GWASs) enable the discovery of genetic variants (e.g., single nucleotide polymorphisms, SNPs) associated with disease outcomes. In brain disease research, the use of imaging quantitative traits (iQTs) as the outcomes provides greater statistical power than using categorical disease status. Recent advancements in neuroimaging and high-throughput sequencing technologies have accelerated the progress of brain imaging genetics, a field that explores the genetic underpinnings of brain structure and function. A number of iQTs GWAS have been conducted, uncovering numerous markers susceptible to brain imaging phenotypes [1,2,3]. However, individual iQT loci (iQTLs) often explain only a small portion of the variance and are challenging to interpret in isolation. To address this, network-based GWAS (NetWAS) methods have emerged, integrating GWAS data with a protein–protein interaction (PPI) network to examine the collective effect of genetic modules on targeted traits.

Several NetWAS approaches have been developed to explore brain biomarkers from the module perspective [4,5]. For example, Yao et al. [6] proposed a tissue-specific NetWAS framework that integrates the brain-specific PPI network to re-prioritize iQT GWAS findings and subsequently identify densely connected and AD-enriched modules as candidates. Meng et al. [7] applied the integrative protein interaction network-based pathway analysis (iPINBPA [8]) on eight subcortical iQT GWAS results and reported modules that were enriched by brain disorders related biological processes. While most existing methods focus on identifying iQT-relevant modules, they often overlook the relationships between these modules and disease. Network-based machine learning approaches have also been proposed, primarily utilizing graph network models to integrate molecular network information for disease prediction. For example, Wang et al. [9] proposed a hypergraph-regularized graph diffusion framework to predict disease outcomes by constructing sample similarity graphs based on integrated brain imaging and genetics data. Wu et al. [10] used multigraph clustering to link genetic variants with brain iQTs in AD, focusing on identifying genetic modules directly. However, the underlying mechanisms connecting iQT-related modules and brain disease, as well as the role of brain tissue in mediating the relationship between genetic modules and disease outcomes, remain largely underexplored.

To bridge the gap between genetic factors and brain diseases, we introduce a novel module-level polygenic risk scores-based (MPRS-based) NetWAS framework to illustrate the pathological paths linking iQT-associated genetic variants to disease and apply it to the study of AD. Our framework integrates genotypes, brain imaging phenotypes, AD clinical diagnosis data, and the PPI network to identify AD-relevant modules driven by iQT-associated variants. In summary, the framework involves the following steps: (1) constructing an amygdala-specific PPI network by integrating GWAS results on amygdala volume with a general PPI network; (2) applying network propagation to identify amygdala-associated gene modules; (3) calculating the MRPS for AD by incorporating AD meta-GWAS data for each module; and (4) keeping modules that show significant differentiation of AD, referred to as AD-relevant modules (ADMs). We further conduct extensive post hoc analyses (e.g., tissue-specificity evaluation, time-to-AD conversion analysis, sensitivity to AD staging, mediation analysis, pathway enrichment analysis) to validate the functional relevance of the identified modules from the perspectives of brain tissue specificity, diagnostic associations, and mediation effects. We demonstrated that the identified ADMs contribute to AD progression through the mediation of the amygdala, exhibit enhanced tissue specificity in the amygdala region, display increased sensitivity to early AD stages, and are enriched in AD-relevant pathways.

## 2. Results

### 2.1. GWAS and GSA of Amygdala QTs

The GWAS analysis investigates associations between 5,574,300 SNPs and grey matter densities of the left and right amygdala, identifying 27 significant associations between 20 SNPs and these two measures. After mapping the SNPs to genes within a ±20 kb window using hg19 genome annotation, MAGMA evaluates associations between 18,093 genes and amygdala QTs, identifying six significant associations (p<(0.05/18,093)=2.76$\;\times \;10^{-6}$; Bonferroni correction) involving three genes (*APOE*, *APOC1*, *TOMM40*) and the two amygdala measures. Detailed results and discussions of the GWAS and GSA are presented in Appendix A.

### 2.2. Network Propagation Identified Amygdala Modules

#### 2.2.1. The Identified Amygdala Modules

The unweighted base PPI network, comprising 185,465 edges among 16,111 gene nodes, was obtained from iRefIndex. After aligning with the amygdala GWAS results, 11,119 genes and 81,576 edges were included in the network propagation analysis. Using the Hierarchical HotNet algorithm, a total of 196 and 195 significant modules (p<0.05, permutation test) were identified from the left and right amygdala-specific networks, respectively. The sizes of amygdala modules range from 3 to 105.

#### 2.2.2. Modularity and Tissue Specificity of Amygdala Modules

We evaluated the modularity and tissue specificity of the identified amygdala modules. As shown in Figure 1a, the modules derived from both the left and right amygdala exhibit significantly higher co-expression than randomly generated modules (*t*-test, p<$1\times 10^{-10}$), highlighting their strong biological relevance and coherence. Figure 1b further demonstrates that these modules show significantly higher co-expression in the amygdala compared to other brain regions or blood, validating their conserved amygdala-specific tissue characteristics.

### 2.3. MPRS Identified ADMs from Amygdala Modules

To identify ADMs from the candidate amygdala modules, we assessed their association with AD by calculating the MPRS for AD and testing for significant differences between the AD and CN groups. By combining results from two meta-GWAS studies, a total of four modules demonstrating significant AD relevance (Bonferroni corrected p<0.05/(196+195)=$1.28\times 10^{-4}$) were identified as ADMs, two from the left amygdala, labeled L.M1 (N=59, $P_{\mathrm{Jansen}}=$$9.86\times 10^{-21}$, $P_{\mathrm{IGAP}}=$$5.13\times 10^{-14}$, Cohen’sd= 0.486, 95% CI: [0.374, 0.598]) and L.M2 (N=16, $P_{\mathrm{Jansen}}=$$1.28\times 10^{-14}$, $P_{\mathrm{IGAP}}=$$1.17\times 10^{-4}$, Cohen’sd= 0.408, 95% CI: [0.296, 0.519]), and two from the right amygdala, labeled R.M3 (N=24, $P_{\mathrm{Jansen}}=$$8.25\times 10^{-22}$, $P_{\mathrm{IGAP}}=$$1.91\times 10^{-16}$, Cohen’sd= 0.494, 95% CI: [0.382, 0.607]) and R.M4 (N=5, $P_{\mathrm{Jansen}}=$$5.31\times 10^{-14}$, $P_{\mathrm{IGAP}}=$$5.22\times 10^{-4}$, Cohen’sd= 0.397, 95% CI: [0.285, 0.509]).

Figure 2a visualizes the four identified ADMs, with node color representing the GSA $-\log_{10}\left(p\right)$ values. Across these modules, a total of 92 unique genes were identified. Among them, 12 genes (*APOE*, *C1RL*, *CD163*, *CD226*, *HBB*, *HBG2*, *HP*, *NEFM*, *PDCD4*, *PLTP*, *PVRL2*, *SELT*) were present in two modules, while the remaining 80 genes were specific to individual modules. Notably, some hub genes have previously been reported as AD risk factors, such as *APOE* [11,12,13,14], *PVRL2* [15,16,17].

### 2.4. ADMs Evaluation and Annotation

#### 2.4.1. ADMs Show Significant Modularity than Random Modules

We assessed the modularity of ADMs by comparing their gene co-expression levels to those of randomly generated modules of the same size. As illustrated in Figure 2b, all four identified modules exhibit significantly higher modularity than random counterparts. Specifically, we performed 1000 permutations for each module, randomly selecting gene sets of the same size from the background network and computing their average co-expression. The results showed that the observed modularity scores of all four ADMs were significantly higher than those of the random modules (permutation p< 0.05 for all modules, *N* = 1000). This suggests that the identified modules are not likely to arise by chance and may capture meaningful functional or regulatory relationships among AD-associated genes.

#### 2.4.2. ADMs Are Sensitivity to AD Progression

We demonstrated that the ADMs identified in this study outperform amygdala iQTs in distinguishing different stages of AD, underscoring their potential as more sensitive and effective biomarkers. Figure 2c illustrates the distribution of MRPS derived from the ADMs and amygdala measurements across various AD stages, and Figure 2d highlights the statistical significance of group comparisons. While left and right amygdala iQTs can distinguish AD from other groups (e.g., the left amygdala shows significance in differentiating AD from CN, EMCI, and LMCI, as well as in EMCI vs. AD; the right amygdala is significant in CN vs. LMCI and LMCI vs. AD), the MPRS derived from ADMs demonstrates particular effectiveness in detecting early AD progression. For example, L.M1 and R.M3 show significant differences between HC and EMCI, and all four modules differentiate EMCI from LMCI. These results underscore the superior sensitivity of ADM-derived MPRS in capturing subtle changes during the early stages of AD, making them more effective biomarkers for tracking disease progression.

Figure 3a,b presents the correlation between ADM-derived MPRS and MCI-to-AD conversion. As shown in Figure 3a, the KM curves for MPRS from each ADM reveal clear stratification between MCI individuals who progress to AD and those who remain stable, with statistically significant differences in survival probabilities across groups (log-rank test, p< 0.05). The multivariate Cox regression, as shown in Figure 3b, further confirms that the MPRS from all four ADMs is a strong predictor of progression risk, with hazard ratios (ranging from 1.256 to 1.387) indicating a robust association between higher MPRS and an increased likelihood of MCI-to-AD conversion.

#### 2.4.3. Amygdala Mediates ADMs to AD

The mediation analysis confirmed that the amygdala is a significant mediator in linking ADMs to AD. As shown in Figure 3c, the amygdala iQT mediates 21.85%, 23.36%, 22.38%, and 18.45% of the effect of MPRS in modules L-M1, L-M3, R-M5, and R-M67, respectively, on AD diagnosis (95% CI, p< 0.001). This consistent mediation across identified modules underscores the significant contribution of amygdala structural alterations to AD pathogenesis, driven in part by genetic influences. These results suggest that the amygdala is not only central to the disease process but also represents a promising target for diagnostic and therapeutic interventions in AD.

#### 2.4.4. Functional Annotation of ADMs

Pathway enrichment analysis was conducted using the Gene Ontology (GO), Kyoto Encyclopedia of Genes and Genomes (KEGG), and Disease Ontology (DO) databases to investigate the functional relevance of the identified modules. Significantly enriched pathways, determined based on an adjusted false discovery rate $P_{\mathrm{FDR}}<$ 0.05, were detailed in Appendix A. We further visualized the enrichment results in Figure 4. For KEGG, all significant terms are included in the visualization. For GO and DO, due to the large number of significant terms enriched, only the six terms with the smallest *p*-values per module were presented. In addition, we visualized the gene pathway network to provide a clearer overview of the module–pathway relationships. An example network illustrating module L.M1 with GO BP terms is presented in Figure 4, with additional plots available in Appendix A.

In the context of AD, several enriched pathways identified through GO, DO, and KEGG analyses highlight their crucial roles in disease pathogenesis. Under the GO terms, pathways such as organelle localization by membrane tethering (GO:0140056) and vesicle docking involved in exocytosis (GO:0006904) are linked to neurodegenerative diseases [18,19], including AD. Alterations in these pathways, particularly in the L.M1 module, have been associated with major psychiatric disorders [20], suggesting a potential mechanistic overlap with AD development.

Within the DO terms, AD (DOID:10652) is explicitly enriched by the module R.M3. The progression of AD has also been linked to vascular conditions such as coronary artery disease (DOID:3393), emphasizing the role of cardiovascular health in AD pathology [14,21,22]. These vascular factors, evident in the pathways enriched by L.M1 and R.M3, act as modifiable risk factors. Additionally, vestibular loss (DOID:3426) which is enriched by the module L.M2, has been reported has been reported to contribute to declining spatial cognitive abilities, a hallmark feature of AD [23].

The KEGG pathway analysis underscores the significance of lipid metabolism in AD, particularly in the module R.M3 [24,25]. Cholesterol levels influence the enzymatic production of beta-amyloid peptide, a key component of AD pathology [26]. Pathways such as SNARE interactions in vesicular transport (hsa04130) and cholesterol metabolism (hsa04979) highlight the critical role of lipid transport and metabolism [20,26], linking genetic predispositions to the neuropathological manifestations of the disease.

Furthermore, the functional enrichment of module R.M4 highlights the role of immune system pathways in AD. Natural killer (NK) cell-mediated processes [27] and alterations in cell adhesion molecules [28,29,30] are noteworthy. These findings suggest intricate interactions between immune dysregulation and AD progression, reinforcing the multifaceted nature of the disease and its underlying mechanisms.

## 3. Discussion

Growing evidence highlights a significant heritable component in the pathogenesis of AD. Concurrently, advancements in brain imaging technologies have enabled the detection of structural and functional changes in specific brain regions that precede the onset of clinical symptoms, underscoring their potential as early indicators of AD. Given the crucial role these brain regions play in AD progression, it is essential to explore how genetic variations influence their structure and function and, in turn, contribute to AD symptoms and disease trajectory. A comprehensive understanding of these brain-related genetic factors holds the potential to refine disease stratification criteria, identify therapeutic targets, and inform strategies for early intervention. Among these regions, the amygdala—crucial for emotional processing and memory—has been consistently implicated in AD pathology. Significant structural and functional alterations in the amygdala are well-documented, and imaging studies have leveraged these changes to aid in the clinical diagnosis of AD.

Building on these insights, we developed a network-based framework to identify genetic modules that influence AD through the functional mediation of the amygdala. Using this approach, we identified four genetic modules associated with AD, thereby demonstrating the utility of our framework in uncovering genetic variants linked to the disease via amygdala-related mechanisms. This framework provides a promising direction for exploring genetic contributions to AD and their imaging mediators, offering valuable insights for future research and clinical applications.

Traditional imaging genomics research primarily focuses on identifying genes or genetic variants related to specific brain regions, often isolating these genetic factors from disease contexts. In our study, by integrating iQT risk factors with AD pathology, four ADMs were identified that are not only associated with the amygdala but also significantly linked to AD risk. Mediation analysis reveals that the amygdala established a crucial link between genetic modules and AD pathology, demonstrating that genetic markers affecting imaging regions can ultimately influence disease progression through the mediation of these imaging regions. Key module members such as *APOE*, *HP*, *KCTD17*, and *PVR* in ADMs, were highlighted for their critical roles in amygdala and AD.

*APOE* acts as a hub gene in the L.M1 and L.M2 modules. Allelic variation in *APOE*, particularly the increased frequency of the ϵ4 allele on chromosome 19, is a major risk factor for late-onset AD [31]. Several studies have reported the impact of the *APOE*ϵ4 allele on amygdala volume [11,32]. The influence of the *APOE* genotype on the amygdala is primarily observed in young-old patients with MCI and AD, with the most significant effects seen in the basolateral, centromedial, and lateral nuclei of the right amygdala, while the basomedial nucleus is less affected [11]. Clusterin (CLU) is associated with tissue damage and the aging process. According to Markowitsch et al. [33], CLU may compromise the structural integrity of the amygdala. Additionally, Den et al. [34] reported that the degeneration of the amygdala in individuals without dementia is linked to an increased risk of developing AD. Therefore, CLU may damage the amygdala, and the degeneration of the amygdala could heighten the risk of developing AD. GWAS and GSA indicate that encoded by rs429358, The *APOE* ϵ4 allele, markedly escalates the lifetime risk of developing AD, significantly impacting both the risk and the age of onset [12,35].

Notably, in modules L.M1 and L.M2, *PDCD4* is directly linked to *APOE* and exhibits a high heat score. Previous studies have identified a connection between *APOE* and *PDCD4*, highlighting their intertwined roles in AD pathology [36,37]. *APOE* modulates Aβ aggregation and clearance, while *PDCD4*, which is upregulated in AD brain tissue, regulates neuronal death, and suppresses protein translation, thereby amplifying Aβ-induced neurotoxicity [36]. These findings position *APOE* and *PDCD4* at the core of a molecular network that may underlie AD development through the amygdala. PDCD4 is a pivotal protein that promotes cell apoptosis within the amygdala, contributing to neuronal death in AD. miR-21 can inhibit apoptosis induced by $A\beta_{1-42}$ by modulating PDCD4 [38], thereby enhancing neuronal survival and affecting synaptic plasticity in the amygdala. Additionally, miR-21 influences the storage and retrieval of emotional memories in the amygdala, with markedly elevated expression of pre-miR-21 suggesting altered miRNA processing or stabilization, potentially mediated by TGF−β1 signaling [39,40,41,42].

In module LM.2, *APOE* is directly linked to the *HP* gene. Previous research has identified associations between the *HP* gene and various cognitive phenotypes [43,44], as well as its physical interactions with *APOE* [45,46] *HP* polymorphisms have been shown to significantly influence AD risk in individuals of European descent [44]. Meta-analyses that include populations of African descent further support this association, demonstrating that *HP* variants modify both the protective effects of *APOE* ϵ2 and the adverse impacts of *APOE*ϵ4 [44]. This modulatory effect is particularly pronounced among carriers of the *APOE*ϵ4 allele. HP may also influence the interaction between *APOE* and Aβ. Spagnuolo et al. [47] demonstrated that HP facilitates the formation of stable Aβ-APOE complexes. Shi et al. [48] reported that APOE and HP, along with four other proteins, are consistently associated with elevated amyloid burden. Similarly, Esiri et al. [49] found that *HP* aids in the binding process between *APOE* and Aβ.

In module RM.3, *KCTD17* functions as a hub gene. *KCTD17* encodes a protein belonging to the potassium channel tetramerization domain-containing family, which has been implicated in both neurodegenerative and psychiatric disorders [50]. Within the module, *KCTD17* and PVRL2 are directly connected, with *PVRL2* previously identified as a risk factor for AD [15,16,17]. The mRNA expression of *KCTD17* in the amygdala increases progressively throughout human brain development, underscoring its role in the maturation of this region [51]. This elevated expression suggests that *KCTD17* is involved in the neural processes underlying the amygdala’s emotional and cognitive functions, thereby indicating that *KCTD17* may influence AD through its effects on the amygdala.

The *PVRL2* has been previously associated with AD and the amygdala in module RM.3. Within module RM.4, *PVRL2* is directly connected to *PVR*, which acts as a key member of the module. *PVR* has been implicated in AD in previous studies [52,53]. The rs10426401 SNP in the first intron of *PVR*, identified as a strong eQTL in AD-relevant tissues, may influence transcription factor binding [54]. Although the association between *PVR* and the amygdala has not yet been validated, a study have reported shared genetic contributions to AD between *TOMM40* and *PVR* in the hippocampus [55]. *TOMM40* was identified through both GWAS and GSA analyses, with the rs59007384 SNP located within the *TOMM40* being associated with the progression from MCI to AD [13]. Furthermore, *TOMM40* has been specifically linked to right amygdala volume [56]. These findings suggest that *PVR* may also influence AD through its effects on the amygdala.

Besides these key member genes, interactions with peripheral genes also highlight the role of the amygdala in the mediating of AD. For example, in LM.1, *INSIG1* is associated with AD by regulating cholesterol homeostasis, which in turn affects the expression of AD risk genes such as *BACE*, *PSEN*, and *APP* [57]. *SCAP*, which is directly connected to *INSIG1*, has been identified as an AD risk factor. Reduction of SCAP suppresses cholesterol synthesis in the brain, leading to impaired synaptic transmission, disrupted long-term potentiation (LTP), and altered cognitive function, which contributes to the increased cognitive decline observed in diabetic states [58]. Furthermore, disruptions in insulin/IGF-1 signaling within the central amygdala, regions potentially influenced by *INSIG1* [59], are linked to metabolic and behavioral abnormalities. These findings suggest that the combined actions of *INSIG1* and *SCAP* may play a crucial role in the mediation of the amygdala of the pathogenesis of AD.

In addition, our analyses provide further evidence that genetic variants influence AD progression through iQT. Four ADMs exhibit high tissue-specific expression in the amygdala and demonstrate significant modularity, thereby aggravating AD progression at the module level. Furthermore, modules L.M1 and R.M3 show heightened sensitivity in differentiating between CN and EMCI states. These findings indicate that biomarkers identified through imaging (amygdala) have superior potential for the early diagnosis of AD.

Future work could expand in several directions. One direction involves integrating longitudinal data analysis to chronologically track AD progression, providing a dynamic perspective on how genetic variations in imaging regions influence the development and progression of AD. Additionally, this approach could be applied to other brain disorders and tissues to identify tissue-specific genetic pathways underlying complex neurological conditions.

This study has several limitations. First, the lack of clinical validation using laboratory and clinical data limits the direct applicability of the identified module biomarkers in clinical settings. Future studies should incorporate clinical datasets to further assess the prognostic value and clinical relevance of these findings. Second, although two independent AD meta-GWAS datasets were used for validation and three significant modules were successfully replicated, the genotyping data for the construction of the MPRS model were exclusively sourced from the ADNI cohort. This restriction may introduce potential cohort-specific biases that affect the generalizability of the results. To ensure the robustness and reproducibility of the proposed approach, independent validation using large-scale genotyping and phenotype data from diverse populations is required. Third, while the identified ADMs demonstrated statistically significant associations with AD ($P_{corrected}<$
$1.28\;\times \;10^{-4}$), the observed effect sizes (Cohen’sd ranging from 0.397 to 0.494) indicate moderate differences in MPRS between AD and CN groups. These consistent effect sizes suggest that the identified ADMs capture meaningful disease-related variations. However, further studies incorporating additional clinical and molecular data are needed to enhance their discriminatory power and evaluate their translational potential in disease classification.

## 4. Methods

To illustrate the effectiveness of the proposed framework for identifying iQT-mediated disease risk modules, we apply it to amygdala imaging genetics in the context of AD. The amygdala is closely linked to AD due to its key role in disease pathology and progression [11,56,60]. Compared to other brain regions such as the hippocampus and cortex, the amygdala not only exhibits early and prominent atrophy in AD but also shows strong genetic associations with emotional memory and neurodegeneration [32,34]. Recent imaging genetics studies suggest that amygdala alterations may precede hippocampus volume loss in certain AD subtypes [6], making it a sensitive and mechanistically informative intermediate phenotype for evaluating genetically mediated disease risk. In addition, the amygdala plays a central role in emotional regulation and affective symptoms (e.g., apathy, anxiety), which are increasingly recognized as prodromal indicators of AD and are themselves heritable traits [61,62]. These characteristics make the amygdala a biologically and clinically relevant intermediate phenotype for dissecting genetically mediated AD mechanisms.

### 4.1. Amygdala-Specific Module Identification

#### 4.1.1. Amygdala-Specific GWAS and Gene-Set Analysis

Neuroimaging and genotyping data for the amygdala-specific GWAS were obtained from the Alzheimer’s Disease Neuroimaging Initiative (ADNI) database. In brief, preprocessed baseline magnetic resonance imaging (MRI) scans and genotyping data were downloaded from the LONI website on 11 June 2021 (adni.loni.usc.edu) and processed as described in [3]. Voxel-based morphometry (VBM) analysis was applied to the MRI data, with the grey matter densities of the left and right amygdala extracted using the MarsBaR AAL atlas. Genotyping data were quality-controlled, imputed, and combined as described in [63]. In total, 5,574,300 SNPs were included in the GWAS. Participants were restricted to non-Hispanic Caucasians to reduce the likelihood of population stratification effects in the genetic analysis. Finally, a total of 1515 subjects were studied, including 353 cognitively normal (CN), 89 with significant memory concern (SMC), 272 with early mild cognitive impairment (EMCI), 508 with late mild cognitive impairment (LMCI), and 293 with AD. Participant characteristics are shown in Table 1. This study was approved by institutional review boards of all participating institutions and written informed consent was obtained from all participants or authorized representatives. See Appendix A for detailed genotyping data and MRI acquisition and processing.

GWAS was performed to assess the associations between amygdala measurements and each SNP, using a linear regression model in PLINK v1.90 [64], adjusting for age, gender, education, and the top four principal components from population stratification analysis as covariates. Since the subsequent network analysis is at the gene level, we employed MAGMA [65] to derive gene-level *p*-values from the GWAS results.

#### 4.1.2. Amygdala-Specific Functional Network Construction

We constructed amygdala-specific genome-wide functional interaction networks by mapping the gene-level *p*-values of amygdala measurements to a general PPI network. Specifically, the base PPI network was sourced from iRefIndex [66], a comprehensive meta-resource aggregating data from more than ten primary interaction databases. Two amygdala-specific networks (left and right) were constructed by assigning gene-level $-\log_{10}\left(p\right)$ values from the corresponding iQT.

#### 4.1.3. Network Propagation to Identify Amygdala Modules

To identify functional modules relevant to the amygdala, we applied network propagation on the amygdala-specific PPI network using the Hierarchical HotNet method [67], as illustrated in Figure 5c. In brief, given the node-weighted interaction network, a joint similarity matrix was derived by taking both the network topology and node weights into the calculation:(1)$$S=\beta {\left(I-\left(1-\beta \right)AD^{-1}\right)}^{-1}\cdot \mathrm{diag}\left(w\left(v_{1}\right),\ldots ,w\left(v_{n}\right)\right),$$
where *A* is the adjacency matrix, *D* is the diagonal degree matrix, β is the random walk restart probability, and $w\left(v_{1}\right),\ldots ,w\left(v_{n}\right)$ are the node weights. Here, the asymmetry of matrix *S* allows it to represent relationships that may be inherently non-symmetrical. Asymmetric hierarchical clustering was then performed, utilizing strongly connected components (SCCs) [68] to construct the dendrogram of gene nodes. Permutation tests were applied to assess the statistical significance of each SCC, determining whether the observed SCCs represented key structural features within the network hierarchy. In our experiments, statistical tests were conducted with 800 permutations, and a minimum network size of 3 was set. See Appendix A for more details about the amygdala module extraction.

### 4.2. MPRS-Based ADMs Identification

To bridge the gap between genetic factors and brain disease with imaging as a mediator, we introduced the module-level polygenic risk score (MPRS), which integrates genotyping data and meta-GWAS statistics of AD for each amygdala module. Modules that exhibited significantly different MPRS between AD and control groups were identified as AD-relevant modules (ADMs). Below, we describe the details of the MPRS-based ADM identification strategy.

#### 4.2.1. Module-Level Polygenic Risk Score

We extended the PRS to the module level to calculate the collective effects of SNPs within each module. We defined the MPRS for an amygdala module $M_{i}$ as the weighted sum of SNPs across all genes within the module, with SNP weights derived from the large-scale AD meta-GWAS:(2)$$\mathrm{MPRS}\left(M_{i}\right)=\sum_{j=1}^{|M_{i}|}\sum_{k=1}^{|g_{j}|}\beta_{j,k}\cdot {\mathrm{SNP}}_{j,k},$$
where $|M_{i}|$ is the number of genes in the module, $|g_{j}|$ represents the number of SNPs within each module gene, ${\mathrm{SNP}}_{j,k}$ is the genotype value of the *k*-th SNP in the *j*-th gene, and $\beta_{j,k}$ denotes the effect size of the corresponding SNP as reported in the AD meta-GWAS.

#### 4.2.2. ADMs Identification

To identify ADMs from the amygdala modules, we assessed the ability of each module to differentiate AD by applying the MPRS to ADNI participants (see Section 4.1.1). Two meta-GWAS studies were used as reference weights, including an AD meta-GWAS (stage 1; 21,982 cases and 41,944 controls) conducted by the International Genomics of Alzheimer’s Project (IGAP) [25] and a large genome-wide meta-analysis of clinically diagnosed AD and AD-by-proxy (phase 3; 71,880 cases and 383,378 controls) conducted by Jansen et al. [69]. In practice, SNPs located within ±20 kb of the module genes were included in the MPRS calculation. The independent *t*-test was used to evaluate the statistical differences in MPRS between the AD and CN groups. Modules with significant differences (Bonferroni corrected p<0.05) are reported as ADMs. Here, we combined significant modules from two meta-GWAS analyses to ensure comprehensive candidate identification.

### 4.3. ADMs Evaluation and Annotation

We hypothesize that the identified ADMs will exhibit amygdala specificity and AD relevance and play a crucial role in linking genetics, brain tissue, and AD. To validate these, we conducted comprehensive experiments to demonstrate the effectiveness of these identified ADMs.

#### 4.3.1. Modularity and Tissue Specificity

To validate the modularity and tissue specificity of the identified modules, we (1) assessed whether the modules display significant co-expression compared to random modules and (2) tested if these modules exhibit higher co-expression in the amygdala relative to other brain and non-brain tissues. Specifically, we used the tissue-specific gene expression data from the Genotype-Tissue Expression (GTEx) project on 1 July 2024 (https://gtexportal.org/), a comprehensive resource to study human gene expression and regulation across diverse tissues and individuals.

Evaluation of amygdala modules. To assess the modularity of the identified amygdala modules, we leveraged the Pearson correlation coefficient (PCC) to measure the co-expression of genes within the module. For each module, the module-level PCC was calculated as the average PCC of all gene pairs within the module. GTEx amygdala expression data were used for PCC calculation, and the paired *t*-test was performed to compare the co-expression level of amygdala modules to those calculated from randomly generated modules of the same size. To ensure the robustness of the comparison, we generated 1000 random modules for each amygdala module and obtained their average module-level PCC for evaluation.

To validate the tissue specificity of the amygdala modules, we compared their co-expression levels in the amygdala with those in other tissues, including the hippocampus, brain cortex, and whole blood. This evaluation covers cortical, subcortical, and non-brain tissues for a comprehensive comparison. A paired *T*-test was conducted to assess the statistical significance of co-expression specificity in the amygdala compared to other tissues. See Appendix A for more details about the evaluation experiments.

Evaluation of ADMs. We further assessed the modularity of each ADM by comparing its average PCC to that of randomly selected modules. For each ADM, we generated 1000 random modules of the same size and conducted a permutation test to assess whether the ADM exhibits significantly higher modularity. We used the GTEx amygdala expression data to calculate the module-level PCC. More details are described in Appendix A.

#### 4.3.2. Sensitivity to AD Progression

To validate the effectiveness of identified ADMs in AD risk assessment, we compared the MPRS across diagnostic groups to analyze differences in genetic susceptibility. A *T*-test was used to compare the distribution and mean MPRS values between groups, examining whether significant differences exist.

Additionally, we performed a time-to-conversion analysis to evaluate the correlation between MPRS and AD progression. More specifically, ADNI participants diagnosed with MCI (including both EMCI and LMCI) at baseline were grouped into progressive (who progressed to AD during the study period) and non-progressive (who did not progress throughout the study period). Conversion time was measured monthly. For the progressive group, conversion duration was determined by measuring the time from initial MCI diagnosis to AD onset. For non-progressive cases, data were right-censored at the last follow-up, reflecting time in MCI without conversion, while AD onset timing remains unknown. Participants were divided into high- and low-risk groups based on median MPRS values. The Kaplan–Meier (KM) method estimated survival probabilities, and the log-rank test assessed statistical significance between groups, allowing us to evaluate whether MPRS-based stratification effectively differentiates levels of genetic risk related to MCI-to-AD conversion. The Cox proportional hazards model assessed MPRS impact on MCI-to-AD conversion risk, estimating hazard ratios (HRs) for MPRS as a continuous variable and between high- and low-risk groups, controlling for confounders (age, gender, and education) to quantify the independent effect of MPRS on AD progression.

#### 4.3.3. Mediation Analysis

To investigate the role of brain imaging in linking genetics to clinical diagnosis, we performed a mediation analysis to assess the mediating effect of the amygdala in the pathological pathway from ADMs to AD. We used the R “mediation” package [70] to construct the x−m−y mediation model and estimate the mediation effect. Here, x represents the independent variable, defined as the MPRS derived from an ADM; y is the dependent variable, representing AD diagnostic status (coded as 0 for CN and SMC, and 1 for MCI and AD); and m is the intermediate variable, defined as the amygdala phenotype, and z represents covariates (age, gender, and education), The mediation analysis followed three steps: (1) a logistic regression model was fitted to assess the effect of x on y, controlling for z; (2) a linear model was fitted to evaluate the effect of x on m, controlling for z; (3) a non-parametric bootstrap approach with 5000 simulations was applied to estimate the direct effect, mediation effect and the proportion of total effect attributable to mediation, and to calculate 95% confidence intervals (CIs) and *p*-values.

#### 4.3.4. Functional Annotation

To evaluate both the functional and AD-specific relevance of genes within the identified ADMs, we performed pathway enrichment analysis using Gene Ontology Biological Processes (GO-BP), Disease Ontology (DO), and KEGG pathways with the R “ClusterProfiler” package [71].

## 5. Conclusions

In this study, we proposed an MPRS-based NetWAS framework to elucidate the genetic mechanisms underlying brain disorders by identifying module-level genetic biomarkers through the mediation of iQTs. Our framework integrates genotypes, brain imaging phenotypes, AD clinical diagnosis data, and the PPI network to identify AD-relevant modules (ADMs) driven by iQT-associated genetic variants. Using amygdala volume as an iQT, we constructed an amygdala-specific PPI network, identified amygdala-associated gene modules through network propagation, and evaluated their AD relevance via MPRS-based differentiation analysis. Extensive post hoc analyses demonstrated that the identified ADMs contribute to AD progression through the mediation of the amygdala, exhibit strong tissue specificity in the amygdala region, display increased sensitivity to early AD stages, and are enriched in AD-related biological pathways. These findings provide novel insights into the genetic architecture of AD and highlight the potential of iQT-mediated and module-level analyses in uncovering disease-related mechanisms.

## Figures and Tables

**Figure 1 ijms-26-06060-f001:**
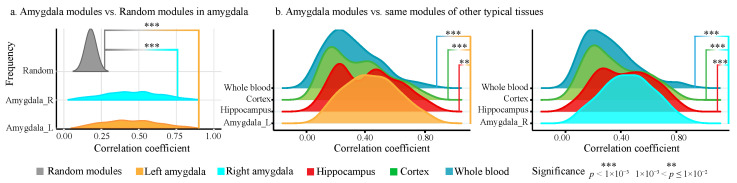
Evaluation of the identified amygdala modules. (**a**) Modules derived from the left and right amygdala exhibit significantly higher co-expression than randomly generated ones (*t*-test, p<$1\times 10^{-10}$). (**b**) The amygdala modules demonstrate higher co-expression in the left and amygdala compared to other brain regions or blood (*T*-test, p<$1\times 10^{-2}$).

**Figure 2 ijms-26-06060-f002:**
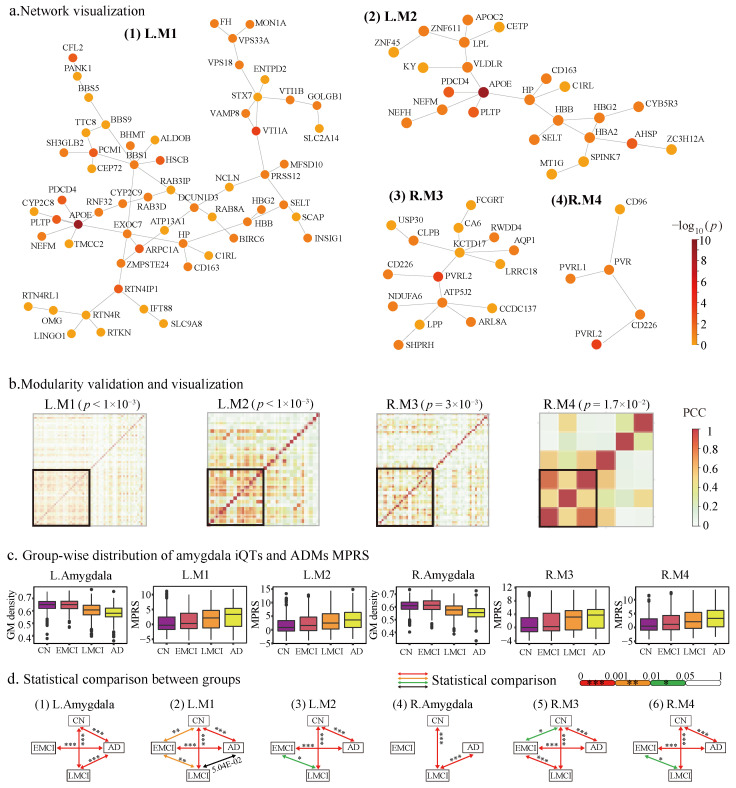
Visualization and evaluation of the identified ADMs. (**a**) Network visualization of the four identified ADMs. Nodes are colored by their gene-level $-\log_{10}\left(p\right)$ values. (**b**) ADMs show significantly higher co-expression patterns compared to random modules. For each ADM, the *p*-value is calculated based on a permutation test with N=1000. The heatmaps illustrate the co-expression patterns of each ADM alongside an example random module of the same size. (**c**,**d**) The sensitivity of ADMs in differentiating AD diagnostic groups is evaluated and compared with amygdala iQTs. (**c**) Boxplots illustrate the distribution of iQTs and ADM-derived MRPS across various AD stages. (**d**) Inter-group analysis reveals that ADMs significantly differentiate AD diagnostic groups more effectively than amygdala iQTs using independent *t*-tests.

**Figure 3 ijms-26-06060-f003:**
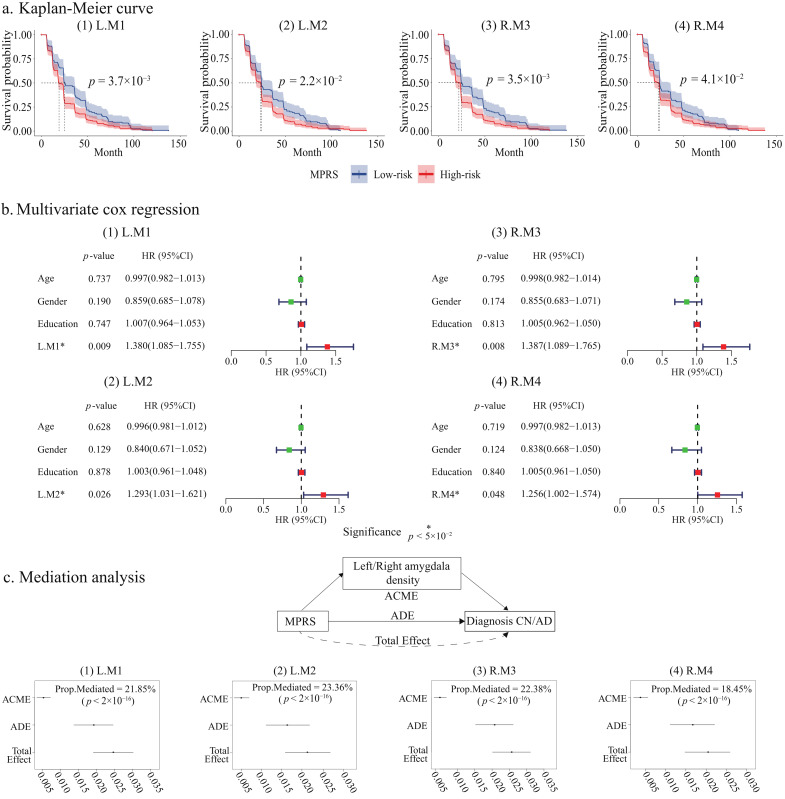
MCI-to-AD conversion analysis and amygdala mediation analysis. (**a**) Kaplan–Meier curves for MCI-to-AD conversion based on ADM-derived MPRS. High-risk and low-risk groups, divided by median MPRS, show significantly different conversion probabilities. (**b**) Hazard ratios from multivariate Cox proportional hazards models, adjusted for age, gender, and education, indicate significant associations between MPRS and AD conversion risk for all four modules. (**c**) Mediation analysis of ADMs on AD through the amygdala. The total effect, average direct effect (ADE), average causal mediation effect (ACME), and mediation proportion are shown for four ADMs. The amygdala iQTs significantly mediate ADMs to AD (p<
$2\times 10^{-16}$).

**Figure 4 ijms-26-06060-f004:**
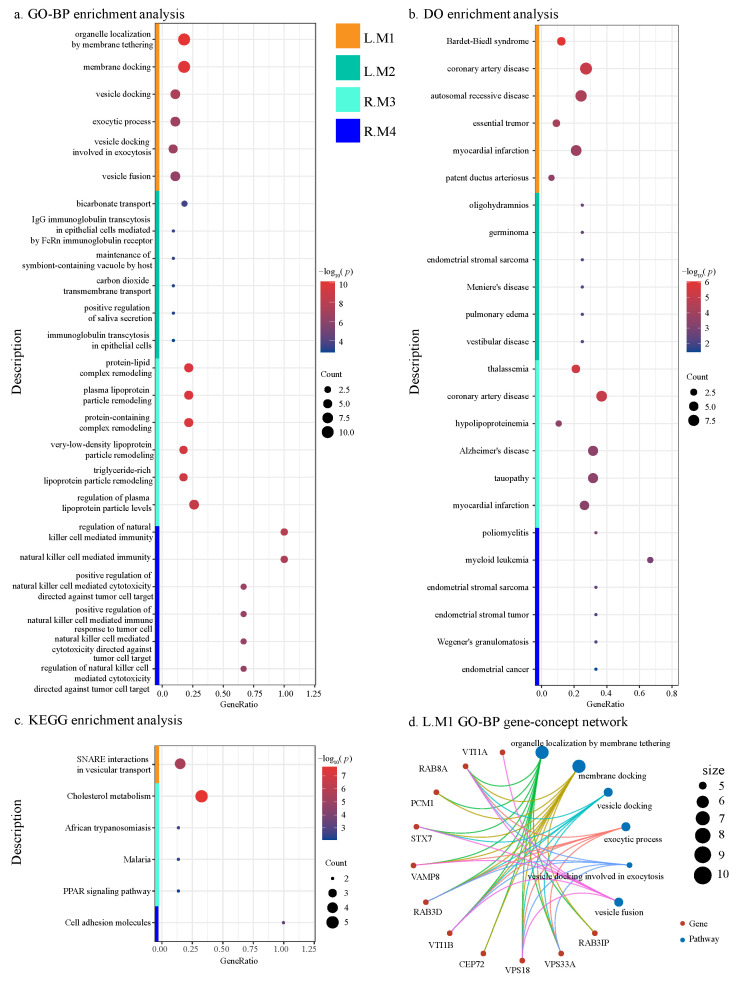
Functional enrichment analysis of the four ADMs. The pathway results from (**a**) GO biological process (GO-BP), (**b**) Disease Ontology (DO), and (**c**) KEGG databases are associated with AD pathology. (**d**) Gene pathway network for module L.M1 with GO-BP.

**Figure 5 ijms-26-06060-f005:**
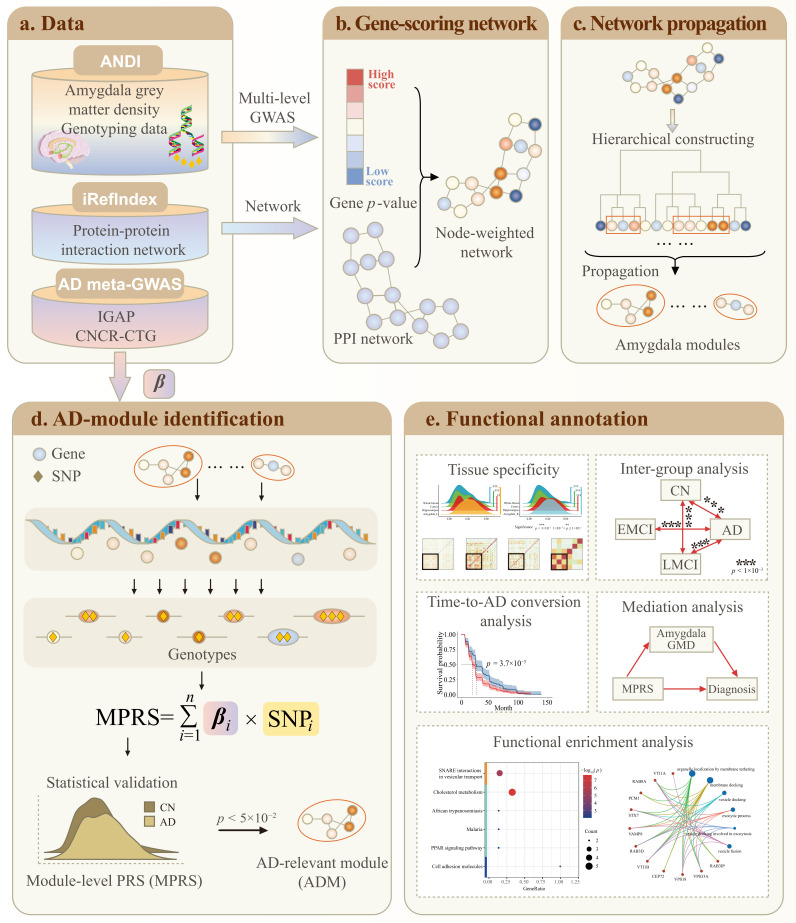
Illustration of the framework. (**a**) Amygdala grey matter density (GMD) and genotyping data from the ADNI cohort are used to conduct GWAS analysis. Tissue-free PPI network is obtained from the iRefIndex database. Summary statistics from two AD meta-GWAS are collected for AD-module identification. (**b**) Amygdala-specific PPI network is constructed by assigning GWAS results of amygdala GMD to the iRefIndex PPI network. (**c**) Amygdala modules are identified through hierarchical network analysis of the amygdala-specific PPI network. (**d**) Module-level polygenic risk score (MPRS) is calculated for each amygdala module by integrating the ADNI genotyping data and AD meta-GWAS statistics. Modules that exhibit significantly different MPRS between AD and control groups were identified as AD-relevant modules (ADMs). (**e**) Functional annotation of the ADMs evaluates their specificity to the amygdala, sensitivity to AD and its progression, pathological pathway via the mediation effect of the amygdala, and the functional relevance of module genes.

**Table 1 ijms-26-06060-t001:** Participants characteristics in the amygdala-specific GWAS.

Diagnosis	CN (*N* = 353)	SMC (*N* = 89)	EMCI (*N* = 272)	LMCI (*N* = 508)	AD (*N* = 293)	*p*-Value
Age	75.14 ± 5.47	72.18 ± 5.73	71.27 ± 7.15	74.06 ± 7.53	75.17 ± 7.90	$<\;\;1\times 10^{-3}$
Gender (M/F)	187/166	36/53	152/120	312/196	164/129	$3\times 10^{-3}$
Education	16.34 ± 2.64	16.76 ± 2.62	16.07 ± 2.64	15.97 ± 2.89	15.18 ± 2.99	$<\;\;1\times 10^{-3}$
Left amygdala GMD	0.64 ± 0.05	0.66 ± 0.04	0.64 ± 0.05	0.60 ± 0.06	0.58 ± 0.06	$<\;\;1\times 10^{-3}$
Right amygdala GMD	0.60 ± 0.05	0.63 ± 0.04	0.61 ± 0.05	0.57 ± 0.05	0.55 ± 0.05	$<\;\;1\times 10^{-3}$

GMD: grey matter density; CN: cognitively normal; SMC: significant memory concern; EMCI: early mild cognitive impairment; LMCI: late mild cognitive impairment; AD: Alzheimer’s disease. Mean and standard deviations are presented in the table. The ANOVA test was used to evaluate differences among diagnostic groups, with the F-test applied for data with homogeneous variances and Welch’s test applied for data with heterogeneous variances.

## Data Availability

The data from the Alzheimer’s Disease Neuroimaging Initiative (ADNI) can be accessed at https://adni.loni.usc.edu/ on 11 June 2021. Meta GWAS data are available from the studies [25,69]. The GTEx dataset can be accessed at https://gtexportal.org/home/ on 1 July 2024.

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
