# Peer review of "A Module-Level Polygenic Risk Score-Based NetWAS Framework for Identifying AD Genetic Modules Mediated by Amygdala: An ADNI Study"

_ijms, 2025, doi:10.3390/ijms26136060_

Round 1

Reviewer 1 Report

Comments and Suggestions for Authors

In the manuscript titled “A module-level polygenic risk score-based NetWAS framework for identifying AD genetic modules mediated by amygdala: an ADNI study,” the authors present a compelling integrative framework that combines genotype data, brain imaging phenotypes, clinical diagnosis of Alzheimer’s Disease (AD), and protein-protein interaction (PPI) networks to identify AD-relevant modules (ADMs) influenced by intermediate quantitative trait (iQT)-associated genetic variants.

A genome-wide association study (GWAS) of amygdala density (N = 1,515) was conducted to identify variants associated with the imaging quantitative trait, which were subsequently mapped onto a PPI network. Through network propagation, amygdala-specific modules were identified. These modules demonstrated strong tissue specificity within the amygdala and were enriched in AD-relevant biological pathways, underscoring the biological relevance of the proposed approach.

Overall, the manuscript is well-written and systematically explains the methodology and findings. However, I believe that a few minor suggestions could further enhance the clarity and impact of the study. I encourage the authors to consider the following suggestions to improve the quality of the manuscript.

  • There are few typos error. For example: in the page number 6, line number 227 ….it is not gray matter, it is grey matter.

  • Throughout the manuscript, authors focused on the methodology with their representative cartoon pictures. When they are talking about clinical phenotype, they may incorporate some real image makes the paper more interesting (If possible).

  • In the section 3.4.1 of results needs some attentions. Authors may include more statistical number to justify their claims.

  • It would be a minor suggestion (if possible) to the authors as the audience of the manuscript will be from the biological background and they may don’t have mathematical background, author may write it in lay-man language to explain it to them. If authors are targeting data science audience , then they may stick to their usual write up.
Comments on the Quality of English Language

It would be a minor suggestion (if possible) to the authors as the audience of the manuscript will be from the biological background and they may don’t have mathematical background, author may write it in lay-man language to explain it to them. If authors are targeting data science audience , then they may stick to their usual write up.

Reviewer 2 Report

Comments and Suggestions for Authors

Herein, a module-level polygenic risk scores-based (MPRS-based) NetWAS framework was presented to explore the pathological paths linking imaging quantitative traits-associated genetic variants to brain disorders. The present work is very interesting, conducted is both thorough and insightful. The manuscript is well structured and presented makes it accessible to a wide audience. Limitation such as the lack of clinical validation are important. Some comments could be taken into account for improvement:

The authors applied it to Alzheimer’s disease, however they should better clarify the selection of amygdala imaging genetics. Comparison with other brain area;

The suggested disease-relevant modules participate to Alzheimer’s disease progression through the mediation of the amygdala. The authors could highlight this evidence in the Figure.

Most of the nodes in network visualization of the four identified ADMs are in the same color, can the author explain this point.

Statistical significance should be included in Figures
